# Interruption of lactate uptake by inhibiting mitochondrial pyruvate transport unravels direct antitumor and radiosensitizing effects

Cyril Corbet[1], Estelle Bastien[1], Nihed Draoui[1,7], Bastien Doix[1], Lionel Mignion[2], Bénédicte F. Jordan[2], Arnaud Marchand[3], Jean-Christophe Vanherck[3], Patrick Chaltin[3], Olivier Schakman[4], Holger M. Becker[5,8], Olivier Riant[6] & Olivier Feron [1]

Lactate exchange between glycolytic and oxidative cancer cells is proposed to optimize tumor growth. Blocking lactate uptake through monocarboxylate transporter 1 (MCT1) represents an attractive therapeutic strategy but may stimulate glucose consumption by oxidative cancer cells. We report here that inhibition of mitochondrial pyruvate carrier (MPC) activity fulfils the tasks of blocking lactate use while preventing glucose oxidative metabolism. Using in vitro $^{13}$C-glucose and in vivo hyperpolarized $^{13}$C-pyruvate, we identify 7ACC2 as a potent inhibitor of mitochondrial pyruvate transport which consecutively blocks extracellular lactate uptake by promoting intracellular pyruvate accumulation. Also, while in spheroids MCT1 inhibition leads to cytostatic effects, MPC activity inhibition induces cytotoxic effects together with glycolysis stimulation and uncompensated inhibition of mitochondrial respiration. Hypoxia reduction obtained with 7ACC2 is further shown to sensitize tumor xenografts to radiotherapy. This study positions MPC as a control point for lactate metabolism and expands on the anticancer potential of MPC inhibition.

[1] Pole of Pharmacology and Therapeutics (FATH), Institut de Recherche Expérimentale et Clinique (IREC), Université catholique de Louvain, 53 Avenue Mounier B1.53.09, Brussels B-1200, Belgium. [2] Louvain Drug Research Institute, Biomedical Magnetic Resonance Research Group, Université catholique de Louvain, 73 Avenue Mounier, REMA 73.08, Brussels B-1200, Belgium. [3] CISTIM Leuven, Center for Drug Design and Discovery (CD3) KU Leuven, Gaston Geenslaan 2, Heverlee 3001, Belgium. [4] Laboratory of Cell Physiology, Institute of Neuroscience, Université catholique de Louvain, Brussels B-1200, Belgium. [5] Division of Zoology/Membrane Transport, FB Biologie, TU Kaiserslautern, P.O. Box 3049, Kaiserslautern D-67653, Germany. [6] Institute of Condensed Matter and Nanosciences, MOST division, Place Louis Pasteur, Université catholique de Louvain, Louvain-la-Neuve B-1348, Belgium. [7] Present address: Department of Oncology, Laboratory of Angiogenesis and Vascular Metabolism, and Vesalius Research Center, VIB, Herestraat 49 box 912, B-3000 Leuven, Belgium. [8] Present address: Institute of Physiological Chemistry, University of Veterinary Medicine Hannover, Bünteweg 17, Hannover D-30559, Germany. Correspondence and requests for materials should be addressed to C.C. (email: cyril.corbet@uclouvain.be) or to O.F. (email: olivier.feron@uclouvain.be)

Tumor lactate-based metabolic symbiosis describes the win-win interaction between lactate-generating and lactate-consuming cells in solid cancers[1,2]. This concept is known for a while in muscle physiology[3,4]. Fast-twitch muscle fibers are indeed known to generate force at the expense of a high anaerobic glycolytic rate while neighboring slow-twitch fibers take up released and circulating lactate to re-generate pyruvate and fuel the TCA cycle. In tumors, the scarcity of nutrients further reinforces the concept of metabolic symbiosis: limitations in glucose availability at some distance of blood vessels may indeed be partly attenuated by the lesser consumption of glucose by the most oxygenated cancer cells that fuel their metabolism with lactate. Spared glucose may thus diffuse in larger amounts towards the most hypoxic cancer cells highly dependent on glucose availability to survive. Today the occurrence of this symbiotic cell–cell lactate shuttle has been reported in a variety of cancer types including cervix, breast and kidney tumors[5,6]. In some instances, cancer-associated fibroblasts (CAF) were also shown to contribute to the lactate-generating compartment within tumors[7–9] although some evidence for lactate consumption by CAF was also reported[10,11]. More recently, resistance to anti-angiogenic treatments was proven to partly arise from exacerbation of lactate-based symbiosis[12–15] and infusing human non-small-cell lung cancer patients with $^{13}$C-lactate revealed extensive labeling of TCA cycle intermediates[16]. In the latter study and another one using genetically engineered mouse models for lung cancer[17], the contribution of lactate to TCA cycle intermediates was further shown to exceed that of glucose; these studies also validate that blood-borne (and not only tumor-derived) lactate may fuel oxidative cancer cells.

Monocarboxylate transporters MCT1 and MCT4 were proposed to support this metabolic symbiosis driven by lactate exchange[2,7,8,12–14,18]. Although lactate molecules are passively transported with protons through MCT transporters, differences in $K_m$ and transcriptional regulation account for major functional differences[6]. A higher affinity for lactate (vs. pyruvate) makes MCT4 the bona fide transporter of lactate outside glycolytic tumor and tumor-associated cells, a function further supported by hypoxia-inducible factor-1α (HIF-1α)-mediated MCT4 gene upregulation[19]. By contrast, the most ubiquitous MCT1 transporter has a low Km for lactate and may thus mediate the capture of lactate from the extracellular compartment when the gradient is favorable.

The identification of MCTs delineates obvious targets to interfere with lactate symbiosis. Although blocking either MCT1 or MCT4 may similarly interrupt cell–cell lactate shuttle, the preferential location of MCT1-expressing cancer cells at the vicinity of blood vessels makes MCT1 a more accessible pharmacological target (than MCT4 often located in the far reached hypoxic regions). Still, largely unaddressed questions are whether cancer cells that express MCT1 may adapt to (and thus resist) the blockade of this transporter and to which extent lactate is preferred to glucose when both fuels are available. Also, although in vitro setups are more appropriate than in vivo experiments to finely dissect the consequences of inhibiting lactate metabolism, more elaborated models than cancer cell monolayers such as 3D spheroids are avidly needed to recapitulate gradients of nutrients and metabolites. Similarly, to more concretely evaluate the clinical potential of targeting metabolic pathways, pharmacological strategies need to be confronted with the issue of drug distribution within distinct tumor compartments.

In this study, we aim to address the above questions by evaluating the effects of two compounds reported to interfere with lactate uptake, namely the MCT1/2 inhibitor AR-C155858[20] and 7ACC2, an anticancer compound originally reported to block lactate influx but not efflux[21,22]. This work identifies 7ACC2 as an inhibitor of mitochondrial pyruvate transport and shows how the blockade of pyruvate import into mitochondria prevents extracellular lactate uptake as efficiently as a MCT1 inhibitor. Contrary to the latter, inhibition of MPC activity keeps exerting its cytotoxic activity when glucose is present and sensitizes tumors to radiotherapy through local reoxygenation.

## Results

**Distinct profiles of drugs blocking lactate uptake**. We used AR-C155858 and 7ACC2 to determine how the presence of glucose influences attempts to alter lactate-based metabolic symbiosis and whether cancer cells may adapt to this therapeutic strategy. AR-C155858 is an inhibitor of MCT1, a major gatekeeper for lactate influx[20] while 7ACC2 is documented to block lactate influx but not efflux[21,22]. Interestingly, while both compounds inhibited cancer cell growth when lactate was the only available fuel (Fig. 1a), only 7ACC2 exhibited anti-proliferative effects on SiHa cervix cancer cells in the presence of glucose (Fig. 1a); similar findings were obtained in two other MCT1-expressing cancer cells (i.e., FaDu and MCF-7 cancer cells; Supplementary Fig. 1a, b). The difference between the two compounds was even more striking when the effects on mitochondrial respiration were measured (Fig. 1b and Supplementary Fig. 1c). Indeed, both compounds inhibited lactate-dependent $O_2$ consumption rate (OCR) in SiHa cells (see also dose response in Fig. 1c) but in the presence of glucose (with or without lactate), AR-C155858 did not affect OCR whereas 7ACC2 dramatically inhibited it (Fig. 1b). The latter results could not be explained by a differential effect on lactate uptake since both compounds similarly inhibited lactate influx (when lactate was the only fuel; Fig. 1d). The lesser affinity of SiHa for glucose (than for lactate) as an oxidative fuel (see Fig. 1b) was neither involved since the use of FaDu and MCF-7 cells that exhibit a similar capacity of glucose and lactate to support OCR confirmed the differential sensitivity to either compound (Supplementary Fig. 1c). In addition, MCT1 and MCT4 expression were not altered upon either treatment (Supplementary Fig. 1d). Of note, in the presence of glucose, lactate efflux (and not uptake) was observed and both drugs failed to inhibit it (Fig. 1e). 7ACC2 treatment actually stimulated lactate release (Fig. 1e) and led to extracellular acidification (Fig. 1f).

**MCT-independent inhibition of lactate uptake**. To understand how drugs interfering with lactate metabolism could differentially affect glucose metabolism, we examined the effects of 7ACC2 and AR-C155858 in *Xenopus* oocytes transduced to express either MCT1 or MCT4. Interestingly, 7ACC2 failed to block lactate uptake (Fig. 2a) and intracellular acidification normally associated with it (Fig. 2b) in MCT1-expressing oocytes while AR-C155858 inhibited both parameters (Fig. 2a, b). In MCT4-expressing oocytes, neither 7ACC2 nor AR-C155858 did influence lactate and proton fluxes (Fig. 2c, d), confirming that AR-C155858 inhibition of lactate flux is MCT1-specific and proving that 7ACC2 activity is not only MCT1- but also MCT4-independent. Of note, longer incubation times with 7ACC2 also failed to reveal inhibitory effects on lactate flux in MCT1- or MCT4-expressing oocytes (Supplementary Figs. 2a, b).

**Inhibition of pyruvate metabolism by 7ACC2**. To investigate how glucose metabolism could be altered upon blockade of lactate uptake (Fig. 1b, d) independently of MCTs (Fig. 2a) we then performed targeted metabolomics analyses to identify changes in the absolute amounts of intracellular metabolites and tracked the fate of $^{13}$C-glucose by analyzing the distribution of major metabolite isotopomers. We found that in comparison to vehicle treatment, intracellular pyruvate and lactate, as well as 3-PG, were

elevated upon 7ACC2 treatment whereas the amounts of most TCA cycle intermediates, except citrate, remained largely unaffected (Fig. 3a). The relative abundance of M+3 lactate and pyruvate upon 7ACC2 treatment was slightly increased, indicating that glucose remained the major contributing nutrient to both metabolites (Fig. 3b, c). A net increase in glucose uptake (Fig. 3d) was actually measured confirming the origin of the absolute increase in lactate (and pyruvate) (see Fig. 3a); note however that the glucose/lactate ratio was not modified (Fig. 3e). Further analysis of the fate of $^{13}$C-glucose however revealed that glucose poorly contributed to major TCA cycle intermediates including M+2 α-KG (α-ketoglutarate), succinate, fumarate and malate (Fig. 3f). Altogether these data concurred to identify pyruvate metabolism as the target of 7ACC2, suggesting that inhibition of lactate influx is an indirect consequence of intracellular pyruvate accumulation. Three other observations support a deficit in the mitochondrial use of pyruvate in the presence of 7ACC2. First, the observed increase in aspartate (Fig. 3a) is most likely to reflect a deficit in the capacity of pyruvate to fuel the TCA cycle (oxaloacetate unable to combine with acetyl-CoA giving rise to aspartate instead of citrate). Second, increases in the relative abundance of M+3 serine and M+2 glycine (Fig. 3g) represent a direct consequence of an increase in the serine synthesis pathway (one of the many side branches of glycolysis) upstream of the pyruvate accumulation in the cytosol. Third, measurements of OCR showed that 7ACC2 could abrogate pyruvate-induced oxygen consumption in intact cells (Fig. 3h) as well as

pyruvate/malate-dependent respiration in isolated mitochondria (Fig. 3i) but not glutamate/malate and succinate-driven respiration (Fig. 3j). The same tracking experiments of $^{13}$C-glucose upon AR-C155858 failed to reveal differences in either M+3 lactate, pyruvate and serine, or M+2 TCA cycle intermediates (Supplementary Figs. 3a, c).

**Blocking MPC activity increases extracellular lactate pool.** We next aimed to further distinguish between inhibitory effects of 7ACC2 on pyruvate transport through mitochondrial pyruvate carrier (MPC) and on pyruvate metabolism through pyruvate dehydrogenase (PDH) and citrate synthase (CS). The two latter hypotheses could be excluded by documenting the lack of effects of 7ACC2 in two specific enzymatic assays (Supplementary Figs. 4a, b); an effect on PDK was also excluded by looking at the extent of PDH phosphorylation by Western blot (Supplementary Fig. 4c). The use of isolated mitochondria however led us to document (i) dose- and time-dependent inhibitory effects of 7ACC2 on pyruvate uptake (Fig. 4a, b), (ii) an associated reduction in matrix acidification using BCECF as a pH-reporter (Fig. 4c) and (iii) the reversibility of OCR inhibition when the membrane-permeable methylpyruvate was used as a substrate (Fig. 4d). To further prove that inhibition of mitochondrial pyruvate uptake could impact on lactate influx, we used the pharmacological MPC inhibitor UK-5099 (see also Supplementary Fig. 4d) and confirmed the observations made with 7ACC2:

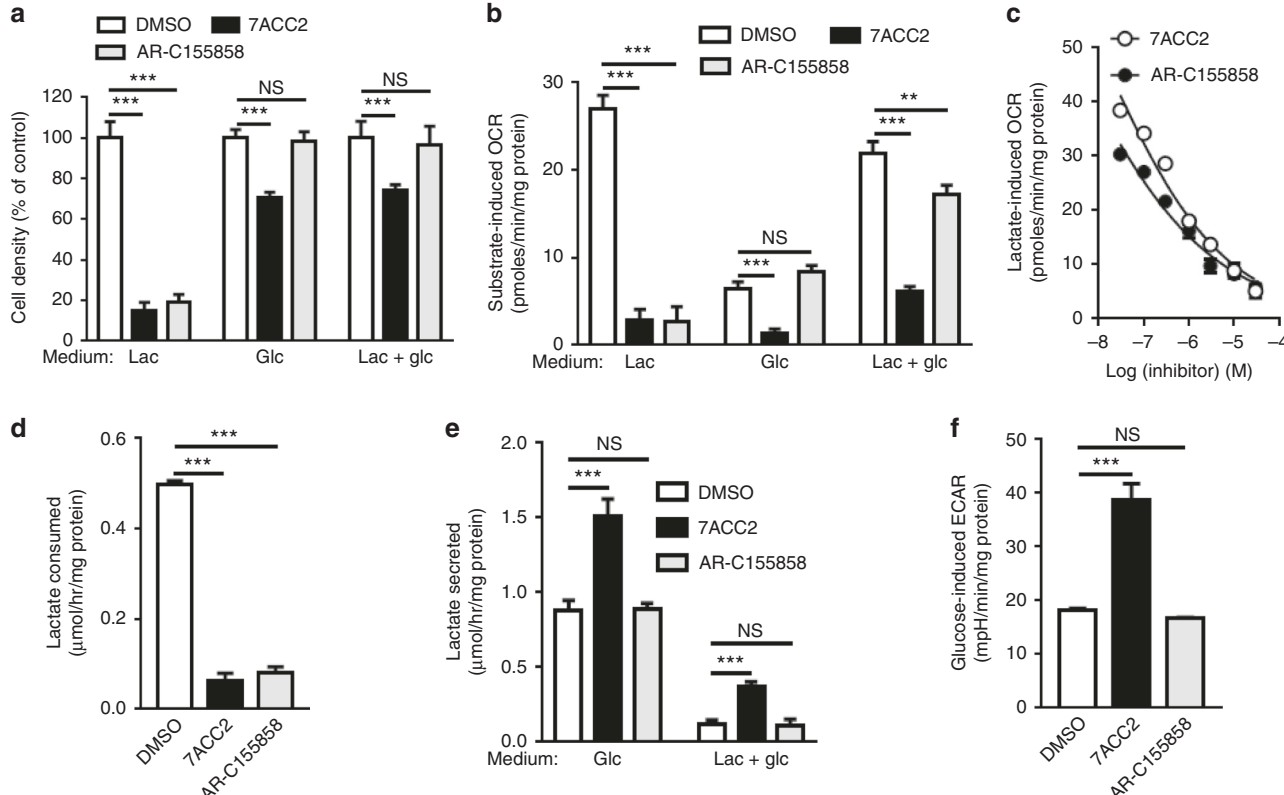

**Fig. 1** Inhibitors of lactate metabolism exert differential effects on cancer cell growth and respiration depending on glucose availability. **a** Cell growth (72 h) and (**b**) oxygen consumption rate (OCR) of SiHa cells after treatment with 10 μM 7ACC2 or AR-C155858 in media containing lactate (Lac) and/or glucose (Glc). **c** Lactate-dependent OCR in SiHa cells exposed to increasing doses of 7ACC2 or AR-C155858. **d** Lactate consumption by SiHa cells incubated for 24 h in a lactate-containing medium. **e** Lactate secretion by SiHa cells incubated for 24 h in indicated media. **f** Extracellular acidification rate (ECAR) of SiHa cells after treatment with 10 μM 7ACC2 or AR-C155858 for 24 h in glucose-containing medium. Data are represented as mean ± SEM of three independent experiments (with ≥6 technical replicates). Significance was determined by one-way (**d**, **f**) or two-way (**a**, **b**, **e**) analysis of variance (ANOVA) with Bonferroni multiple-comparison analysis. **p < 0.01; ***p < 0.001; NS, not significant

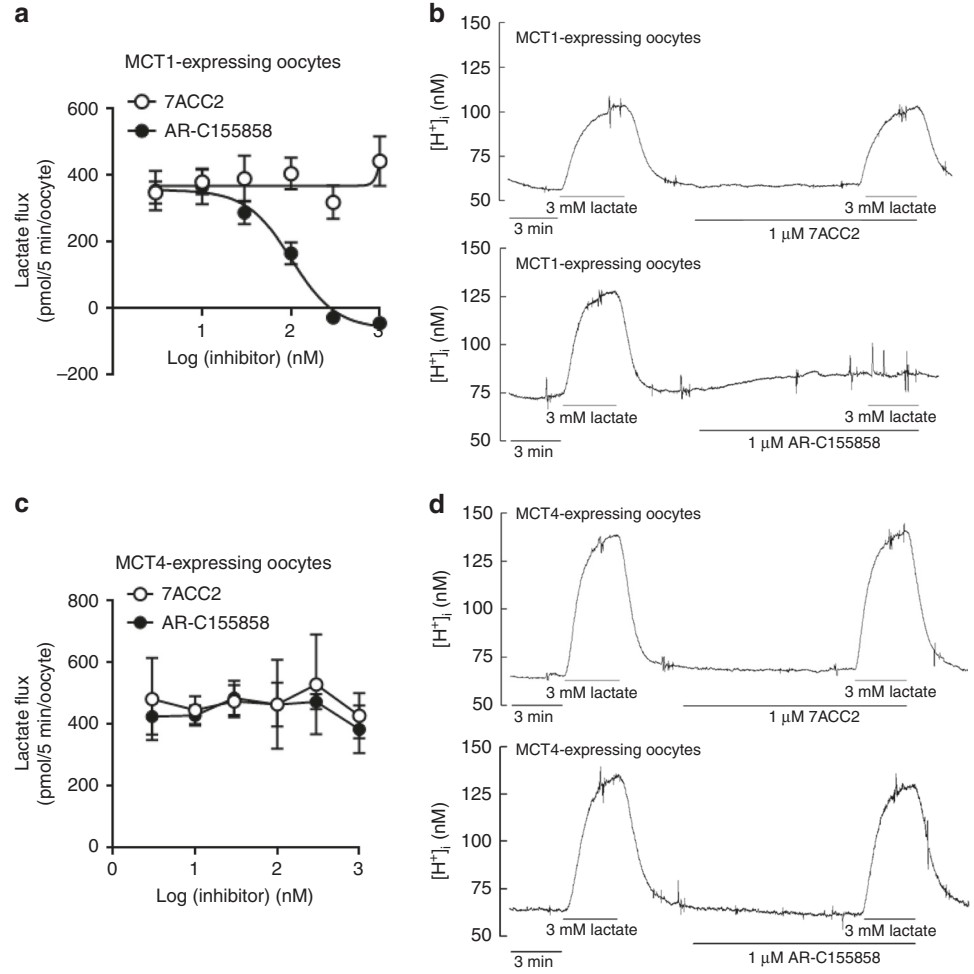

**Fig. 2** AR-C155858 but not 7ACC2 inhibits MCT1 activity. **a** [14]C-lactate uptake in MCT1-expressing *Xenopus* oocytes after treatment with increasing doses of 7ACC2 or AR-C155858 ($n = 8$ for each group). **b** Original recording of intracellular H[+] concentration in MCT1-expressing *Xenopus* oocytes during application and after removal of 3 mM lactate in the absence and presence of 7ACC2 or AR-C155858. **c** [14]C-lactate uptake and (**d**) intracellular H[+] concentration in MCT4-expressing *Xenopus* oocytes in response to 7ACC2 or AR-C155858 treatments ($n = 8$ for each group). Data are represented as mean ± SEM

uptake of extracellular lactate was blocked (Fig. 4e) and in the presence of glucose, an increase in lactate release could be observed (Fig. 4f).

Finally, we used intravenous injection of hyperpolarized [1−13C]pyruvate in tumor-bearing mice and measured signals from [1−13C]pyruvate, and from [1−13C]lactate upon exchange of 13C label between injected pyruvate and endogenous lactate. We found that 7ACC2 led to an increase in 13C-lactate in the tumors (Fig. 4g, h), thereby confirming the effects observed in vitro. By contrast, the MCT inhibitor AR-C155858 favored a decrease in 13C-lactate pool in mouse tumors reflecting the inhibitory effects of this drug on the uptake of lactate but also of pyruvate from the extracellular medium (Fig. 4g, h). As a consequence, the tumor lactate/pyruvate ratio evolved in opposite direction in animals treated with 7ACC2 and AR-C155858 (Fig. 4i). In parallel, we also determined the effects of both drugs on tumor burden and found a significantly larger growth inhibitory effect in mice treated with 7ACC2 (vs. AR-C155858) (Fig. 4j).

It is noteworthy that the pool of 13C-alanine (the other potential anaerobic/cytosolic product of glucose metabolism) was also reduced in the presence of 7ACC2 (Fig. 4g and Supplementary Figs. 4e, f). This observation is in agreement with data obtained in vitro (Fig. 3a and Supplementary Fig. 4g) and

supports a major role of mitochondrial (and not cytosolic) aminotransferase enzyme ALT in the generation of alanine from pyruvate in our experimental models.

**Distinct antitumor effects of 7ACC2 and AR-C155858.** The above findings led us to examine how the differential influence of MPC and MCT1 inhibitors on lactate and pyruvate handling could influence tumor growth in three-dimensional (3D) cell culture models as well as in vivo. Using spheroids obtained from 3D FaDu cell cultures (more prone to generate spheroids than SiHa cells) we found that either MCT or MPC inhibition reduced the growth of tumor spheroids (Fig. 5a, b) but while cytostatic effects were obtained with AR-C155858, inhibition of mitochondrial pyruvate transport by 7ACC2 led to cytotoxic effects. We could indeed observe that in response to 7ACC2 treatment (but not upon AR-C155858 exposure), cellular density within spheroids progressively decreased (Fig. 5c) and a halo of dead cells developed around spheroids (Fig. 5a, d). Propidium iodide staining confirmed that 7ACC2 promoted cell death to a larger extent than AR-C155858 (Supplementary Figs. 5a, b) while Ki-67 staining revealed that the latter exerted more significant antiproliferative effects than upon 7ACC2 exposure (Supplementary Figs. 5c, d). To further characterize metabolic changes induced by

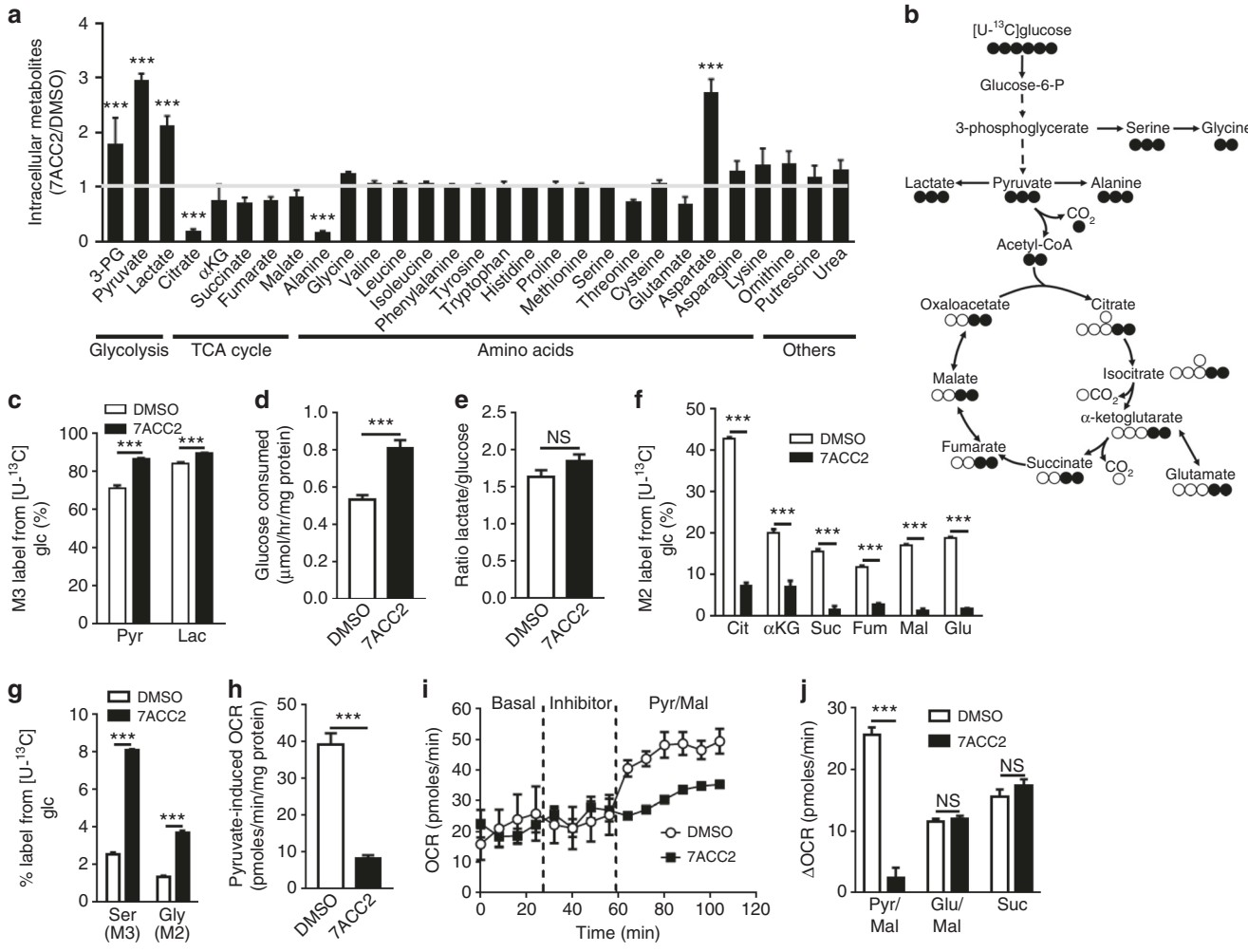

**Fig. 3** Inhibition of lactate uptake by 7ACC2 is associated with increased glycolytic flux and reduced cell respiration. **a** Relative abundance of intracellular metabolites in SiHa cells after treatment with 10 μM 7ACC2 for 24 h. **b** Carbon atom transition map depicting oxidation of [U-$^{13}C_6$]glucose. **c** Relative abundance of glycolysis-derived pyruvate and lactate, (**d**) glucose consumption, (**e**) lactate/glucose ratio, relative abundance of indicated (**f**) TCA cycle intermediates and (**g**) serine synthesis pathway metabolites in SiHa cells treated or not with 10 μM 7ACC2 for 24 h. **h** Pyruvate-dependent OCR in SiHa cells after treatment with 10 μM 7ACC2. **i** OCR measurements using isolated SiHa cell mitochondria in response to pyruvate/malate substrates after treatment with 10 μM 7ACC2 and (**j**) extent of OCR increase in response to indicated substrates. Data are represented as mean ± SEM of three independent experiments (with ≥6 technical replicates). Significance was determined by Student's $t$-test (**d**, **e**, **h**), one-way ANOVA (**a**) or two-way ANOVA (**c**, **f**, **g**, **j**) with Bonferroni multiple-comparison analysis. ***$p < 0.001$; NS, not significant

MPC vs. MCT inhibition, we used IRDye-2-deoxyglucose (2DG-IR), a fluorescent glucose analog to monitor glucose uptake across spheroid cell layers. To avoid the confusing effects of cell death, we exposed spheroids to either treatment for a limited period of 24 h. We found a more diffuse 2DG-IR staining upon 7ACC2 treatment than following AR-C155858 exposure (Fig. 5e, f); quantification confirmed that the extent of glucose analog that reached the spheroid core was not altered by AR-C155858 treatment but was significantly increased after 7ACC2 treatment (Fig. 5g). We also found a higher and more diffuse expression of GLUT-1 in spheroids exposed to 7ACC2 (vs. control and AR-C155858-treated spheroids; Fig. 5h, i), in agreement with the global increase in glucose uptake upon 7ACC2 treatment observed in 2D cultures (see Fig. 3d). An increase in glucose uptake was also observed in spheroids generated from shMPC1-expressing cancer cells (Fig. 5j–l) or exposed to the pharmacological MPC inhibitor UK-5099 (Supplementary Figs. 5e, f). A significant growth inhibitory effect was however only observed in spheroids treated by the UK-5099 compound (Supplementary Fig. 5g; to the same extent as upon 7ACC2 treatment (Fig. 5b)),

suggesting a compensatory mechanism when MPC is completely silenced before the formation of spheroids.

**Tumor radiosensitization upon MPC activity inhibition**. We then reasoned that if more glucose is consumed but that pyruvate cannot fuel mitochondrial respiratory metabolism (as previously validated in 2D cultures (Figs. 1b, c and 3h)), an increase in oxygen availability should be observed. Immunostaining with CA9 and pimonidazole (both used as hypoxia markers) confirmed a net reduction in the extent of hypoxia in spheroids exposed to 7ACC2 (vs. control and AR-C155858-treated spheroids; Fig. 6a). We further examined whether such reoxygenation upon 7ACC2 treatment could be exploited as a radiosensitizing strategy since local increase in tumor oxygenation is well known to increase the efficacy of radiation treatments. We first documented that 7ACC2 in vivo treatment could lead to tumor reoxygenation using electron paramagnetic resonance (EPR) technique (Fig. 6b). We then combined 7ACC2 i.p. administration with radiotherapy administered as a single high dose (16 Gy;

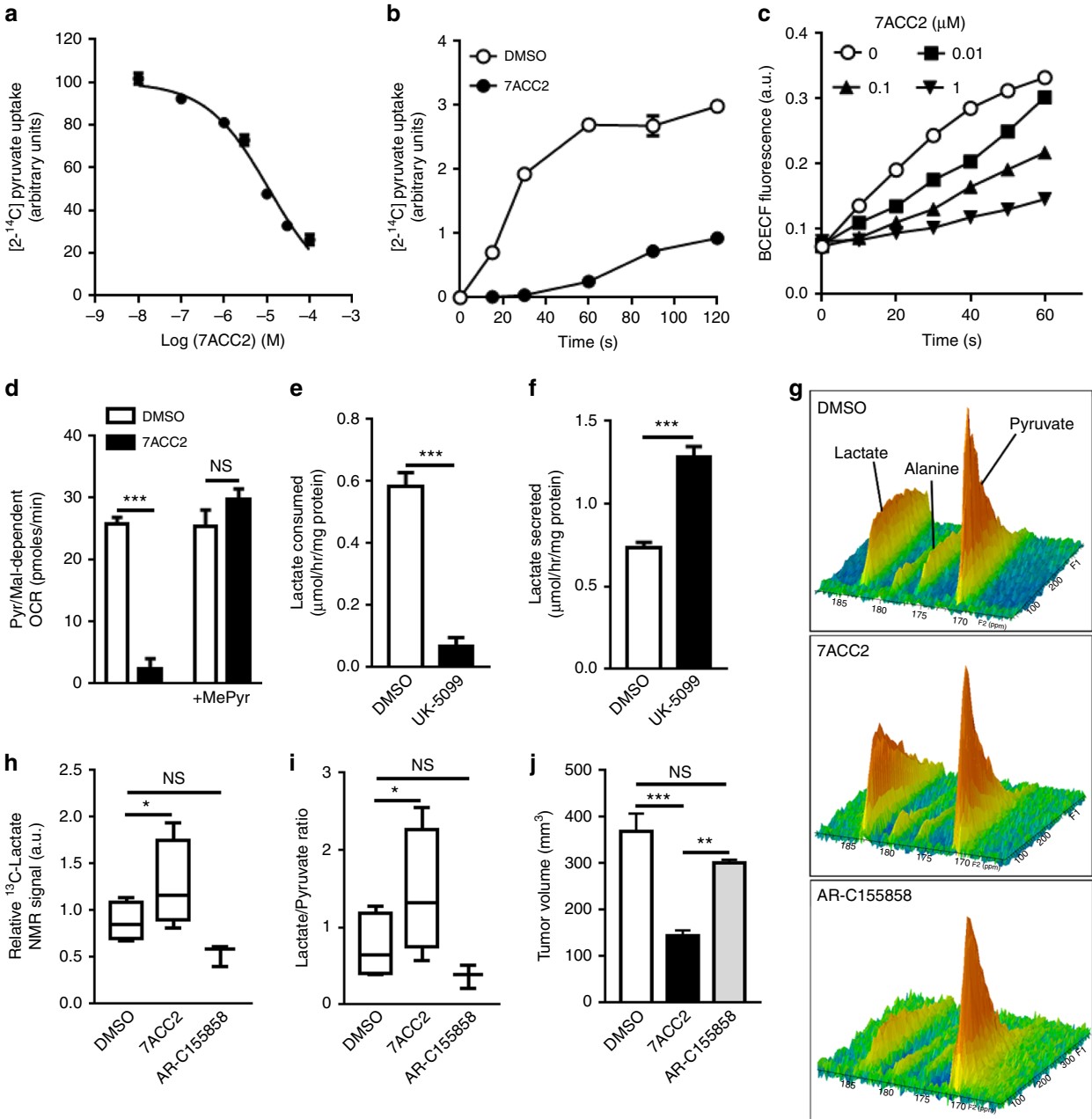

**Fig. 4** 7ACC2 specifically inhibits mitochondrial pyruvate transport. [2–$^{14}$C]pyruvate uptake in isolated mitochondria from SiHa cells treated (**a**) with increasing doses of 7ACC2 for 2 min or (**b**) with 10 μM 7ACC2 for increasing periods of time. **c** BCECF fluorescence reflecting progressive changes in pH within isolated SiHa cell mitochondria treated with increasing doses of 7ACC2. **d** Effects of 10 μM 7ACC2 on OCR measurements using isolated SiHa cell mitochondria exposed to pyruvate/malate with or without methylpyruvate. **e** Lactate consumption and (**f**) secretion by SiHa cells incubated for 24 h with 10 μM UK-5099 in lactate or glucose-containing medium, respectively. **g** Representative $^{13}$C-MRS spectra acquired in vivo from treated and untreated SiHa tumor xenografts after hyperpolarized $^{13}$C-pyruvate injection. **h** Relative $^{13}$C-lactate NMR signals and (**i**) modifications in the $^{13}$C lactate/pyruvate ratio after treatment with 3 mg/kg 7ACC2 or AR-C155858 for 2 h ($n = 3$ for each group). **j** Tumor volume of SiHa xenografts in nude mice after 9-day treatment with 3 mg/kg 7ACC2 or AR-C155858 ($n = 5$ for each group). Data are represented as mean ± SEM of three independent experiments (with ≥ 6 technical replicates). Significance was determined by Student's *t*-test (**e**, **f**), one-way ANOVA (**h–j**) or two-way ANOVA (**d**) with Bonferroni multiple-comparison analysis. *$p < 0.05$; **$p < 0.01$; ***$p < 0.001$; NS, not significant

Fig. 6c) or according to a fractionated mode (5 × 4 Gy; Fig. 6d). In both setups, a significant increase in tumor growth delay was observed by combining 7ACC2 treatment with radiotherapy (Fig. 6c, d), supporting a radiosensitizing effect of MPC inhibition, in particular in the fractionated mode of radiotherapy where either treatment alone barely influenced tumor growth (Fig. 6d).

A significant radiosensitizing effect was also observed in tumors derived from cancer cells expressing two distinct MPC1-targeted shRNA (Supplementary Fig. 6a) as well as upon administration of UK-5099 compound (Supplementary Fig. 6b) together with the demonstration of a reduction in hypoxia as revealed by pimonidazole staining (Supplementary Fig. 6c).

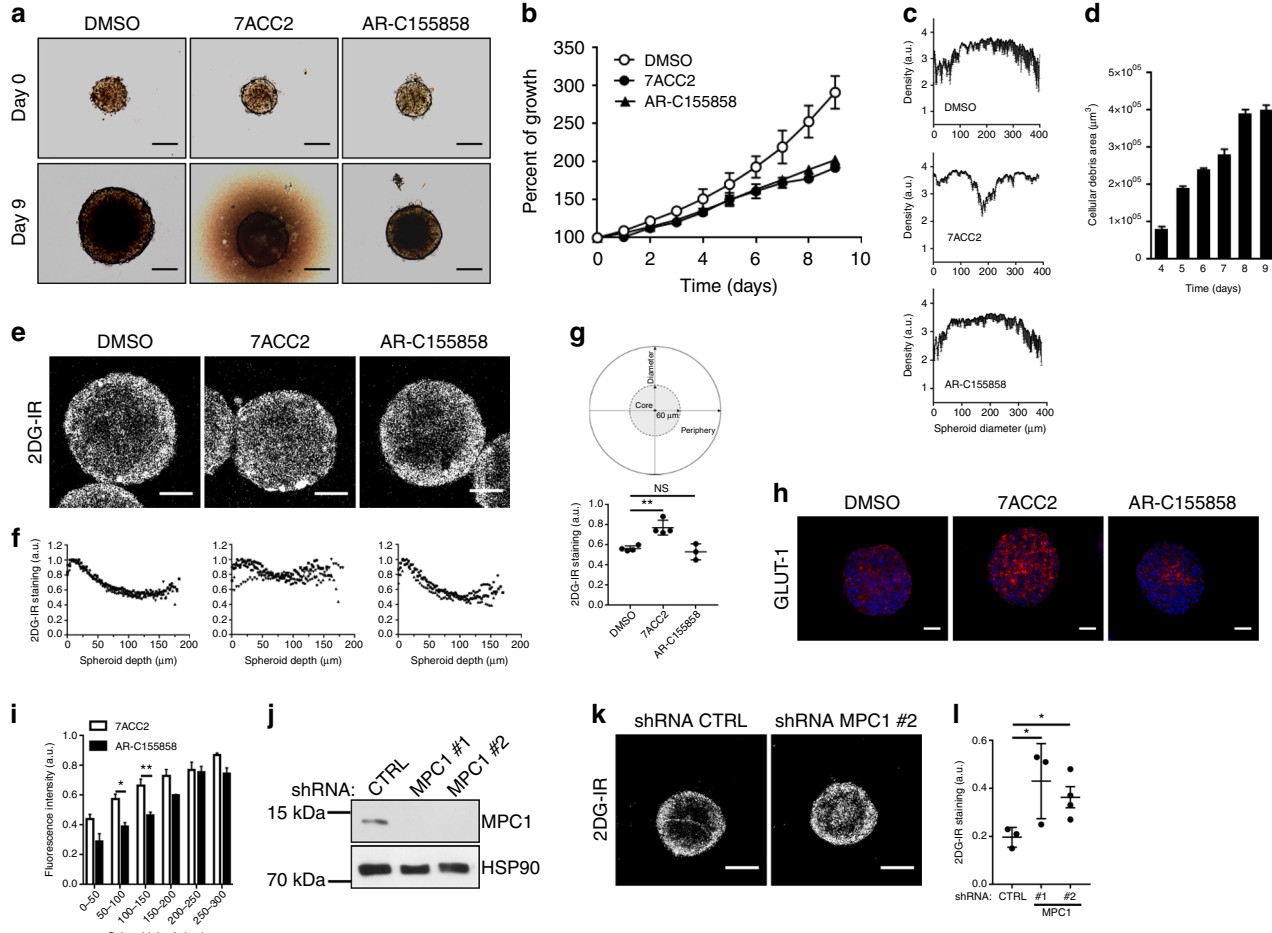

**Fig. 5** 7ACC2 and AR-C155858 reduce the growth of tumor spheroids through different mechanisms. **a** Representative pictures and (**b**) time-dependent growth of FaDu spheroids upon treatment with 20 μM 7ACC2 or AR-C155858 for 9 days. Scale bars: 200 μm. **c** Cellular density within FaDu spheroids treated as indicated in **a** for 9 days. **d** Time-dependent accumulation of cellular debris (i.e., appearance of a brown ring surrounding spheroids) from FaDu spheroids treated with 20 μM 7ACC2. **e** Representative immunofluorescence pictures and quantification of 2-deoxyglucose-IRDye (2DG-IR, 3 h exposure), (**f**) distribution from the periphery to the core and (**g**) accumulation within the core of FaDu spheroids pre-exposed to 20 μM 7ACC2 or AR-C155858 for 24 h. Scale bars: 100 μm. **h** Representative immunofluorescent pictures and (**i**) corresponding quantification (at the indicated spheroid depth) of GLUT-1 staining in FaDu spheroids treated with 20 μM 7ACC2 or AR-C155858 for 5 days. Scale bars: 100 μm. **j** Representative immunoblotting for MPC1 in FaDu cells expressing control or MPC1-targeting shRNA sequences. **k** Representative immunofluorescence pictures and (**l**) quantification of 2DG-IR within the spheroid core (as in **g**) in FaDu spheroids expressing control or MPC1-targeting shRNA. Scale bars: 200 μm. Data are represented as mean ± SEM of three independent experiments (with ≥6 technical replicates). Significance was determined by one-way ANOVA (**g**, **l**) or two-way ANOVA (**i**) with Bonferroni multiple-comparison analysis. *$p < 0.05$; **$p < 0.01$; NS, not significant

## Discussion

Metabolic symbiosis is one expression of tumor flexibility. Cell–cell lactate shuttle is a prototypical example of how the end-product of a metabolic path for some cancer cells may be exploited by other cells to extract the remaining bioenergetic/biosynthetic potential contained in an apparent waste product[1,6]. Interrupting lactate-driven metabolic symbiosis thus represents an attractive therapeutic strategy since it bears the hope to target metabolically distinct compartments with one single drug. In this study, through the discovery of 7ACC2 as a compound endowed with MPC inhibitory activity, we documented that blockade of mitochondrial transport of pyruvate can block the uptake and thus the use of extracellular lactate by cancer cells. The significance of this finding is threefold. First, our data make inhibition of mitochondrial pyruvate transport a much more attractive strategy than MCT blockade to inhibit lactate-driven metabolic symbiosis since it also prevents the (compensatory) use of glucose when the latter is available. Second, the shift from a partially aerobic glucose metabolism to strict anaerobic glycolysis

upon MPC activity inhibition is associated with cytotoxic effects (not observed in response to MCT1 inhibition). This finding is at odd with the view that exacerbated glycolysis is necessarily associated with negative outcomes and needs to be combated[23–26] but in line with recent studies that have identified direct links between mitochondrial activity and cancer progression[27–32]. Third, by blocking TCA cycle fueling by both lactate and glucose, inhibition of mitochondrial pyruvate transport also dramatically reduces oxygen consumption rate and thereby sensitizes tumors to radiotherapy. This indicates that even if other anaplerotic pathways can compensate deficient MPC activity, reduced contributions of glucose and lactate to mitochondrial pyruvate is sufficient to profoundly alter OXPHOS and locally increase $O_2$ availability.

The identification of 7ACC2 as a small molecule capable to block mitochondrial pyruvate transport was guided by the profile of this compound able to inhibit lactate influx, while not altering its efflux and not preventing the uptake of the MCT1 substrate 3-bromopyruvate[22]. The current study demonstrates that inhibition

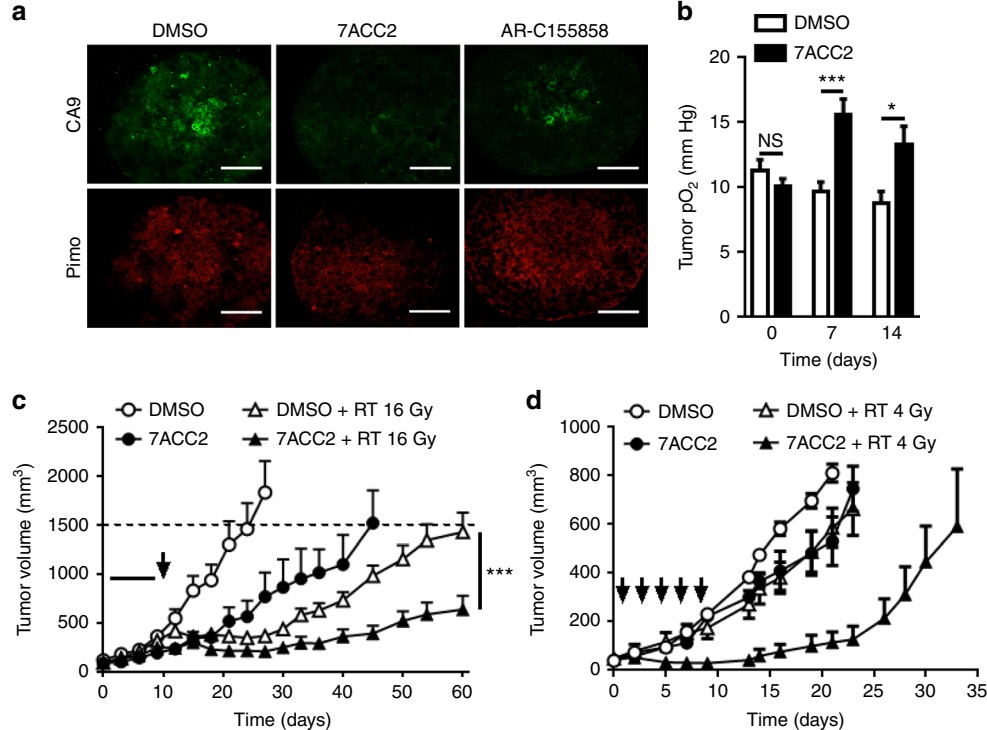

**Fig. 6** Inhibition of mitochondrial pyruvate transport radiosensitizes tumor cells by reducing hypoxia in vivo. **a** Representative immunofluorescent pictures of CA9 and pimonidazole stainings in FaDu spheroids treated with 20 μM 7ACC2 or AR-C155858 for 5 days. Scale bars: 100 μm. **b** Tumor pO$_2$ measurements by EPR oximetry in SiHa xenografts treated with 3 mg/kg 7ACC2 for the indicated period of time ($n = 5$ for each group). **c, d** Tumor growth of SiHa xenografts in nude mice daily treated with 3 mg/kg 7ACC2 (for 10 and 5 days in **c** and **d**, respectively) followed (or not) by a single irradiation of (**c**) 16 Gy or (**d**) 5 fractions of 4 Gy; arrows indicate the days of irradiation ($n = 7$ for each group). 7ACC2 was formulated in DMSO (**c**) or in 5% DMA/50% HPβCD/45% sodium phosphate buffer (**d**). Data are represented as mean ± SEM. Significance was determined by Student's t-test (**c**), or two-way ANOVA (**b**) with Bonferroni multiple-comparison analysis. *$p < 0.05$; ***$p < 0.001$; NS, not significant

of lactate uptake by 7ACC2 actually arises from the intracellular conversion of lactate into pyruvate and the failure to further oxidize it when mitochondria transport is blocked. Cytosolic accumulation of pyruvate then rapidly prevents further conversion of lactate into new pyruvate molecules, which in turn limits lactate influx and thereby interrupts metabolic symbiosis. Our data thus provide evidence that even though 7ACC2 does not bind to MCT1 (or MCT4), it blocks lactate uptake as efficiently as the MCT1 inhibitor AR-C155558.

The combined use of 2D and 3D cultures of cancer cells brought complementary evidence that MPC inhibition may offer further advantages over MCT1 blockade. In conventional 2D cultures, although MCT1 and MPC blockers inhibit cancer cell growth and respiration when lactate is the only nutrient available, the presence of glucose abrogates the effects of MCT1 blockade but not those resulting from the inhibition of mitochondrial pyruvate transport by 7ACC2. These results also indicate that glucose is preferred to lactate when both substrates are available and therefore that MPC inhibition represents a more attractive strategy to interfere with cancer cell oxidative metabolism whatever the nature of available nutrients (i.e., glucose or lactate). Data derived from spheroid models expanded on this observation. Indeed, while both inhibitors blocked the growth of spheroids, cytostatic effects were observed with MCT1 inhibitor AR-C155858 while 7ACC2 showed significant cytotoxic effects. This potent growth inhibitory action is associated with a shift towards a more glycolytic phenotype as revealed by in vitro increases in glucose uptake and lactate efflux and in vivo hyperpolarization NMR data documenting a 7ACC2-driven increase in pyruvate to lactate conversion. Interestingly, these metabolic alterations also

led to a reduction in hypoxia validated in spheroids via pimonidazole and CAIX staining and in vivo through EPR measurements. This led us to unravel the radiosensitizing potential of the inhibition of mitochondrial pyruvate uptake by documenting how 7ACC2 administration considerably improves the efficacy of either single high dose or fractionated low dose radiotherapy.

MPC activity inhibition is not without consequences in healthy tissues and we actually observed exercise intolerance in 7ACC2-treated mice (vs. sham-treated mice) with an elapse time until exhaustion being reduced by 40% (Supplementary Fig. 6d). Although this could represent a potential discomfort for cancer patients, it should however not prevent the use of such drug as long as patients are kept at rest during the treatment. At the dosage used in our study, neither off-targets nor gross tissue toxicity were actually observed with 7ACC2 (Supplementary Note 1). This could represent an advantage regarding two other recently identified inhibitors of mitochondrial pyruvate transport, namely the insulin sensitizers thiazolidinediones[33] and the phosphodiesterase inhibitor zaprinast[34]; the former are known to provoke water retention, increased adiposity and bone loss while headaches, dyspepsia and visual deficits have been reported with the latter. More generally, a potential limitation to the use of these different compounds as anticancer drugs is the deficit in MPC expression in some cancers, in particular colorectal cancers (CRC)[35]. Several studies also reported an association between tumor progression and MPC deficiency[36–38]. Although more work is needed to identify the determinants of MPC expression in tumors, adaptation to MPC inactivation may actually vary according to cell types and experimental protocols. For instance, contrary to our data, Vacanti and colleagues showed that in

C2C12 myotubes MPC inhibition did not alter cell respiration and did not lead to increased glycolysis[38]; excess cytosolic pyruvate in these cells was actually not diverted to lactate or alanine but instead directly secreted. Schell et al. reported that re-expression of MPC1/2 in CRC cells, wherein basal MPC expression is null or very low, led to increased OCR and growth inhibition[37]. This finding may be related to our observation that the growth of spheroids made of MPC1-silenced cancer cells is not inhibited while both 7ACC2 and UK-5099 dramatically prevent the growth of MPC1-expressing spheroids. Finally, Yang and colleagues reported that MPC blockade induced acetyl-CoA formation from glutamine using a pathway confined to the mitochondria to compensate for the lack of mitochondrial glucose-derived pyruvate[36]; whether stimulation of this anaplerotic pathway influenced global cell respiration was however not addressed by the authors. Altogether these studies stress that detection of MPC expression in tumors represents a prerequisite to identify cancers the most likely to respond to a MPC inhibitor, either used for direct anticancer effects or its radiosensitizing potential. Of note, recent studies that have used administration of [13]C-labeled fuels to tumor-bearing mice[17,39,40] but also cancer patients[16,41–43] have provided evidence that besides aerobic glycolysis, [13]C carbons from both glucose and lactate were converted into TCA cycle intermediates, further supporting a role of MPC anaplerosis and other biosynthetic activities.

In conclusion, while at first glance MCTs are the most obvious targets to block lactate-based symbiosis in tumors, we showed here that blocking mitochondrial pyruvate transport leads to a similar inhibition of lactate uptake but at the same time alters glucose oxidation, the two effects adding up to support a direct strong cytotoxic activity but also to sensitize tumors to radiotherapy.

## Methods

**Reagents**. Antimycin A, Ascorbic acid, D-glucose, DL-malic acid, methylpyruvate, N, N, N′, N′-tetramethyl-p-phenylenediamine (TMPD), potassium dichloroacetate, pyruvic acid, rotenone, sodium L-lactate, succinic acid and UK-5099 were from Sigma-Aldrich. [U-[13]C]glucose tracer was from Cambridge Isotope Laboratories. [3]H$_2$O was from American Radiolabeled Chemicals and [14]C tracers were from Perkin Elmer. AR-C155858 was purchased from Tocris Bioscience. 7ACC2 (7-aminocarboxycoumarin 2, see structure in Supplementary Note 1) and CPI-613 were synthesized and purified in our lab.

**Cell culture**. All cell lines were acquired in the last 3 years from ATCC where they are regularly authenticated by short tandem repeat profiling. Cells were stored according to the supplier's instructions and used within 6 months after resuscitation of frozen aliquots. SiHa cervix cancer cells, FaDu pharynx squamous cell carcinoma cells, MCF-7 and MDA-MB-231 breast cancer cells were maintained in DMEM supplemented with 10% heat-inactivated FBS. All cell lines were tested for mycoplasma contamination with the MycoAlert™ Mycoplasma Detection kit (Lonza) before being used. Cell growth was assessed by using the Presto Blue reagent (Life Technologies) according to manufacturer's instructions.

**MPC1 silencing**. Specific shRNA sequences targeting human *MPC1* gene (shRNA MPC1 #1: V3SH11240-226346303; shRNA MPC1 #2: V3SH11240-228593341), expressed in a SMARTvector lentiviral plasmid (Dharmacon), were transfected in FaDu cells by using the lipofectamine LTX reagent (ThermoFisher Scientific) according to the manufacturer's instructions. A non-targeting shRNA sequence (VSC11278; Dharmacon) was used as a negative control. shRNA-expressing cells were selected after treatment with 0.5 μg/ml puromycin for 14 days and silencing efficiency was assessed by Western blotting.

**Metabolic profiling**. Glucose and lactate concentrations were measured using enzymatic assays (CMA Microdialysis AB) and a CMA 600 analyzer (Aurora Borealis). Oxygen consumption rate (OCR) and extracellular acidification rate (ECAR) were measured using the Seahorse XF96 plate reader. Briefly, cells ($3 \times 10^4$ cells/well) were pre-challenged in a substrate-free medium with 7ACC2 or AR-C155858 for 30 min and then treated with 10 mM lactate, glucose or pyruvate as indicated. In specific assays, 4 μg of isolated mitochondria were plated, pre-challenged in a substrate-free medium with 7ACC2 or AR-C155858 for 30 min and then treated with 10 mM pyruvate/malate, glutamate/malate or succinate.

Substrate-dependent OCR or ECAR was calculated by comparing the values after and before addition of the substrates.

**Gas chromatography-mass spectrometry (GC-MS) analysis**. For tracer studies, phenol red-free DMEM was formulated with 10% dialyzed FBS and supplemented with 10 mM D-[U-[13]C]glucose and 2 mM unlabeled L-glutamine. Cultures were washed with PBS before adding tracer media for 24 h. Tumor cells were rinsed in cold 0.9% NaCl and snap-frozen in liquid nitrogen. Mass spectrometry measurements ([13]C isotope analysis and absolute quantification of intracellular metabolites) were then performed by the Metabolism Core in the Sanford Medical Discovery Institute (La Jolla)[44]. All mass spectrometry data and calculations are provided in Supplementary Data 1.

**Mitochondrial pyruvate transport assays**. Mitochondria were isolated from tumor cells as described before[45]. Pyruvate uptake into mitochondria was then measured at 4 °C using an ascorbate/TMPD-generated proton gradient and the inhibitor-stop technique as reported previously[46]. The pH-sensitive fluorescent dye 2′, 7′-bis(carboxyethyl)-5(6)-carboxyfluorescein (BCECF) was also used to monitor the change in matrix pH changes associated with the proton-linked uptake of the pyruvate analog dichloroacetate, as described elsewhere[47].

**Western blot analysis**. Western blotting experiments were carried out as previously described[48]. Anti-HSP90 (610419; BD Biosciences), anti-MPC1 (ab74871; Abcam), anti-phospho-PDHE1α (Ser-293) (ABS204; Millipore) and anti-PDHE1α (#3205; Cell Signaling Technology) antibodies were used at a dilution of 1/1000 (or 1/250 for the anti-MPC1 antibody) in 5% bovine serum albumin (BSA). Anti-MCT1 and anti-MCT4 antibodies (custom-made, ThermoFisher Scientific) were used at 1/1000 in 5% skimmed milk. Uncropped versions of Western blots are presented in Supplementary Fig. 7.

**Enzymatic assays**. Specific activities for citrate synthase (CS) and pyruvate dehydrogenase (PDH) enzymes were assessed by using dedicated enzyme activity assays from Abcam, according to manufacturer's instructions.

**3D cultures**. Spheroids were prepared with FaDu cells by seeding 1500 cells/well in ultra-low attachment 96-well plate (Corning) in DMEM supplemented with 10% heat-inactivated FBS, 10 mM D-glucose and 2 mM L-glutamine. Inhibitors were added on day 4 (mean diameter of spheroids: 300 μm) and renewed every 3 days. Spheroid growth was monitored using live-cell phase contrast microscope (Axio Observer, Zeiss); spheroid area, density and morphology were measured using ImageJ software.

For glucose labeling, spheroids were treated for 24 h with 20 μM 7ACC2, AR-C155858 or UK-5099, then incubated for 30 min in glucose-free medium and finally exposed for 3 h with 50 μM IRDye-800CW-2-deoxyglucose (2DG-IR, LI-COR Biosciences). After two washes in PBS, 2DG-IR uptake was imaged in whole spheroids with a confocal fluorescence microscope LSM510 Zeiss (laser at 633 nm and emission filter LP685 nm). A Z-stacking was performed on each spheroid to select images at 60 μm of depth. A similar protocol was used for propidium iodide staining. Briefly, spheroids were treated for 5 days with 20 μM 7ACC2 or AR-C155858 and then suspended in propidium iodide (PI) solution at 2 mg/ml for 5 min at room temperature. After two washes in PBS, PI staining was measured on whole spheroids with a confocal fluorescence microscope LSM510-Meta Multiphoton Zeiss (laser at 890 nm and bandpass filter 600–750 nm). A Z-stacking was performed on each spheroid to select images at 80 μm of depth.

For immunohistochemical studies, spheroids were treated for 5 days with 20 μM 7ACC2 or AR-C155858, then harvested and embedded in OCT. Frozen sections (5 μm) were fixed in ice-cold acetone and stained with following primary antibodies: CA9 (NB100-417F; Novus Biologicals), Ki67 (556003; BD Biosciences), GLUT-1 (ab652; Abcam). Sections were incubated with anti-mouse/anti-rabbit antibodies coupled to Alexa Fluor, and nuclei were counterstained with DAPI. Slides were prepared with fluorescence mounting medium (Dako), and staining was visualized by Zeiss Imager 1.0 Apotome microscope. All spheroid samples from a same experiment were imaged by using the same gain and exposure settings.

**In vivo tumor xenograft experiments**. All the experiments involving mouse xenograft models received the approval of the ethic committee from the Université Catholique de Louvain (approval ID 2012/UCL/MD005) and were carried out according to national care regulations. 7-week-old female NMRI nude mice were purchased from Elevage Janvier, and $2.10^6$ tumor cells were injected subcutaneously in the left flank of the animals. When tumors reached a mean diameter of 5 mm, 7ACC2, AR-C155858 or UK-5099 (3 mg/kg; resuspended in DMSO) was daily injected intraperitoneally. 7ACC2 was also formulated in 5% N,N-dimethyl-lacetamide/50% hydroxypropyl-β-cyclodextrin/45% sodium phosphate buffer pH 8.0. In some experimental groups, 7ACC2 or UK-5099 treatment was combined with tumor irradiation using a [137]Cs γ-irradiator (IBL-637; CIS-BioInternational) for a total absorbed dose of 4 or 16 Gy.

**Pimonidazole staining on tumor sections**. 7-week-old female NMRI nude mice were purchased from Elevage Janvier, and $2.10^6$ tumor cells were injected subcutaneously in the left flank of the animals. In some conditions, mice were treated for 7 consecutive days with i.p. injections of 5 mg/kg UK-5099. When tumors reached a mean diameter of 7 mm, pimonidazole (60 mg/kg; Hypoxyprobe) was injected i.v. in the tail vein of the mice. Tumors were excised 2 h later, fixed in paraformaldehyde and embedded into paraffin. Tumor sections (5 μm) were then immunostained by using a HRP-conjugated anti-pimonidazole antibody (Hypoxyprobe). Images were acquired by using a slide scanner (SCN400; Leica) and analyzed with the TissueIA software (Leica).

**Hyperpolarized $^{13}$C-NMR spectroscopy and data analysis**. [1-$^{13}$C] pyruvic acid (Sigma-Aldrich) was mixed with 15 mM trityl radical OX63 and doped with 2 mM gadoteric acid (Guerbet). This solution of 40 μl was hyperpolarized by an Oxford DNP Polarizer (HyperSense) for approximately 45 min at 1.4 K and 3.35 T. The polarized substrate was quickly dissolved in 3 ml of heated buffer containing 100 mg/l EDTA, 40 mM HEPES, 30 mM NaCl, and 80 mM NaOH. The final solution was at neutral pH and contained hyperpolarized [1-$^{13}$C] pyruvate and non-hyperpolarized, unlabeled lactate (30 mM) to increase $^{13}$C label exchange[49]. This solution was quickly injected using a catheter into the tail vein of the mice in the MRI scanner (11.7-Tesla, Bruker, Biospec). Mice were anesthetized by isoflurane inhalation. A warm water blanket was used to maintain body temperature and a pressure cushion was used to monitor breathing. Mice were scanned using a double tuned $^1$H-$^{13}$C-surface coil (RAPID Biomedical), which was designed for spectroscopy of subcutaneous tumors (i.e., tumor-shaped cavity of 12 mm in diameter). After administration of 0.3 mL of hyperpolarized pyruvate, $^{13}$C spectra were acquired using a single pulse sequence every 3 sec for 210 sec, a flip angle of 10° and a bandwidth of 12.8 kHz. Peak areas under the curve were measured for each repetition time and each time point using homemade routines in MATLAB (Mathworks). The integrated peak intensities of hyperpolarized $^{13}$C-pyruvate, $^{13}$C-lactate, $^{13}$C-alanine and $^{13}$C global signal were measured. The rate of exchange between pyruvate and alanine and lactate was also calculated.

**EPR oximetry**. In vivo tumor $pO_2$ was monitored using Electron paramagnetic resonance (EPR) oximetry using an EPR spectrometer (Magnettech, Berlin, Germany) with a low-frequency microwave bridge operating at 1.1 GHz and an extended loop resonator. A charcoal (Charcoal wood powder, CX0670-1; EM Science, Gibbstown, NJ, USA) was used as the oxygen-sensitive probe in all experiments. Calibrations curves were made by measuring the EPR line width as a function of the $pO2$.

**Determination of MCT transport activity in Xenopus oocytes**. Plasmid DNA of rat MCT1 (SLC16A1 gene) and rat MCT4 (SLC16A3 gene), cloned into the oocyte expression vector pGEM-He-Juel, which contains the 5′ and the 3′ untranscribed regions of the Xenopus β-globin flanking the multiple cloning site, was transcribed in vitro by T7 RNA-Polymerase (Ambion mMessage mMachine, Life technologies) as described earlier[50,51]. Segments of ovarian lobules were surgically removed under sterile conditions from Xenopus laevis frogs, anaesthetized with 1 g/l of 3-amino-benzoic acid ethylester (MS-222, Sigma-Aldrich) and rendered hypothermic. The procedure was approved by the Landesuntersuchungsamt Rheinland-Pfalz, Koblenz (23 177-07/A07-2-003 §6). Oocytes were singularized by collagenase (Collagenase A, Roche) treatment in $Ca^{2+}$-free oocyte saline (pH 7.8) and stored overnight in $Ca^{2+}$-containing oocyte saline (in mM: 82.5 NaCl, 2.5 KCl, 1 $CaCl_2$, 1 $MgCl_2$, 1 $Na_2HPO_4$, 5 HEPES), supplemented with gentamycin, at 18 °C to recover. Three to five days before the experiment, oocytes of the stages V and VI were injected with 5 ng of cRNA coding for MCT1 or MCT4, using a microinjection system (WPI Nanoliter 2000, World Precision Instruments Germany GmbH).

For determination of $IC_{50}$, groups of eight oocytes were incubated in 95 μl of $Ca^{2+}$-containing oocyte saline (pH 7.4), supplemented with 0, 0.003, 0.01, 0.03, 0.1, 0.3, and 1 μM of inhibitor for 10 min in a 5 ml polypropylene tube. After incubation, 5 μl of uptake solution, containing 0.2 μCi of $^{14}$C-labeled lactate plus unlabeled lactate to yield a final lactate concentration of 3 mM (for MCT1) or 10 mM (for MCT4), respectively, were added. Uptake was stopped after 5 min by washing oocytes four times with 4 ml of ice-cold oocyte saline. For determination of time dependence, groups of 10 oocytes were incubated in 95 μl of $Ca^{2+}$-containing oocyte saline, supplemented with 1 μM of inhibitor for 15 and 45 min, respectively. After incubation, 5 μl of uptake solution, containing 0.6 μCi of $^{14}$C-labeled lactate were added. Uptake was again stopped after 5 min by washing oocytes four times with 4 ml of ice-cold oocyte saline. Single oocytes were placed into scintillation vials and lysed by addition of 200 μl 10% SDS and vortexing. After lysis, 3 ml scintillation fluid (Rotiszint® eco plus scintillation cocktail, Carl Roth) was added and the radioactivity determined by liquid scintillation counting using a Tri-Carb 2810TR scintillation counter (Perkin Elmer). In each experiment, groups of native oocytes were incubated in uptake solution under the same conditions and the measured activity was subtracted from the activity measured in MCT-expressing oocytes.

Changes in intracellular $H^+$ concentration in oocytes were determined with ion-sensitive microelectrodes under voltage-clamp conditions as described in detail previously[50,51]. All measurements were carried out in HEPES-buffered, $Ca^{2+}$-containing oocyte saline (pH 7.0) at room temperature. In lactate-containing saline, NaCl was replaced by an equivalent amount of Na-L-lactate. Inhibitor was added to the oocyte saline shortly before the experiment.

**Forced treadmill exercise**. Exercise experiments received the approval of the ethic committee from the Université Catholique de Louvain (approval ID 2017/UCL/MD/013) and were carried out according to national care regulations. 8-week-old male C57BL/6 N mice were purchased from Elevage Janvier. 2 h after i.p. injection with 7ACC2 (3 mg/kg) or DMSO (vehicle), mice were placed on a homemade treadmill to run with an uphill inclination of 15° at a speed of 5 m/min for 1 min, followed by a progressive increase in speed of 1 m/min every minute up to 18 m/min, as previously described[52]. The back of the treadmill was equipped with a grid that discharged a mild electrical current, a stimulus aimed at motivating the animal to keep running on the treadmill. The test was stopped when the mouse remained on the shocker plate without attempting to reengage the treadmill, and the time to exhaustion was determined.

**Statistical analysis**. Results are expressed as mean ± SEM of at least three independent experiments, unless otherwise noted. Sample sizes (n) are reported in the corresponding figure legends. No pre-specified effect size was calculated, and no statistical method was used to predetermine sample size. None of the samples/animals was excluded from the experiment. For in vivo studies, the investigators were not blinded to allocation during experiments. Mice were allocated on the basis of their pretreatment engraftment levels, and no method of randomization was used. All mice were cared for equally in an unbiased fashion by animal technicians and investigators. Two-tailed unpaired Student t-test, one-way or two-way ANOVA tests (Bonferroni's post-hoc test) were used where appropriate. Where indicated, variables were transformed using the natural logarithms, before t-tests were performed to meet the assumption of equal variances.

**Data availability**. The data supporting the findings of this study are available within the article and the associated Supplementary Information. Any other data is available from the corresponding author upon reasonable request.

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

## Acknowledgements
We thank Céline Guilbaud and Laurenne Petit for their excellent technical assistance. We also thank David Scott and Adam Richardson (Cancer Metabolism Core, La Jolla) for the $^{13}$C metabolomics studies. This work was supported by grants from the Fonds national de la Recherche Scientifique (FRS-FNRS), the Belgian Foundation against cancer, the J. Maisin Foundation, the interuniversity attraction pole (IUAP) research program #UP7-03 from the Belgian Science Policy Office (Belspo) and an Action de Recherche Concertée (ARC 14/19).

## Author contributions
Conceptualization, C.C. and O.F.; Methodology, C.C., E.B., N.D.; B.D., L.M., O.S. and H.M.B.; Investigation, C.C., E.B., N.D.; Writing – Original Draft, C.C. and O.F.; Writing —Review and Editing, C.C. and O.F.; Funding Acquisition, O.F.; Resources, B.D., B.F.J., A.M., J.-C.V., P.C., O.R. and O.S.; Supervision, C.C. and O.F.

## Additional information

**Competing interests:** The authors declare no competing interests.

