## [Peer Review File(PDF 719 kb) · Nature Communications]

Reviewers' comments:

Reviewer #1 (Remarks to the Author):

In this manuscript, utilizing a number of experiments including in vitro metabolomics and in vivo MRI, the authors demonstrate that 7ACC2 is a potent MPC inhibitor that appears to not target MCT1. The authors also find that 7ACC2 reduces tumor spheroid growth and radiosensitizes tumor cells in vivo. The discovery of a new MPC inhibitor is already an interesting development and they do this very well. However, they do not do an adequate job of connecting this drug mechanism data to the phenotypic effects. All drugs have off-target effects and one needs to be very careful when attempting to conclude mechanistic information from phenotypic data. It is possible that 7ACC2 is an MPC inhibitor, which is quite clearly the case, and that its effects on tumor growth are due to another off-target mechanism. Therefore, the following experiments should be conducted:

1. Treat spheroids with UK5099, to see if this pharmacological MPC inhibitor also reduces tumor spheroid growth and induces glucose uptake.
2. Genetically knockout MPC in the tumor spheroids, to determine if MPC depletion also reduces its growth and induces glucose uptake.
3. Treat xenografts in nude mice with UK5099, to test whether MPC inhibition by UK5099 also radiosensitizes tumor cells and causes tumor reoxygenation.
4. Introduce MPC KO xenografts in nude mice, to see if KO MPC radiosensitizes tumor cells and causes tumor reoxygenation as well.

Reviewer #2 (Remarks to the Author):

The present manuscript provides extensive indirect evidence that 7ACC2 inhibits mitochondrial pyruvate transport. It further provides evidence that 7ACC2 can inhibit xenograft growth and sensitize such tumors to radiotherapy. Overall, I found the quality of the evidence good, and the metabolic reasoning straightforward, transparent, and accurate. Particularly pleasing is the use of multiple measurement modalities, at the 2D culture, 3D culture, and xenograft level, to reach generally consistent conclusions.

Suggested improvements before publication:

1. The authors should carefully think through the difference between "mitochondrial pyruvate transport" inhibition (in title, and well shown) vs. "MCP inhibition" which implies a specific molecular target where direct evidence is limited. Wording should be chosen to accurately reflect the nature of the data and to include caveats where data is limited.
2. The first part of the title seems unnecessary. Lactate-based metabolic symbiosis may or may not exist. The contribution of this paper is to show that 7ACC2 blocks mitochondrial pyruvate transport and thereby inhibits tumor growth and sensitizes to radiotherapy.
3. The authors should briefly discuss alternative MCP inhibitors, their state of development, pros and cons, and potency and in vivo utility.
4. The authors should provide the careful documentation of the safety/toxicology of the employed doses of 7ACC2 in vivo. One would anticipate that blocking mitochondrial pyruvate uptake could result in substantial toxicity in almost any organ, perhaps especially the brain or in prolonged exercise. Evidence of blood brain barrier penetration (or not) would be useful, as would exercise studies and tissue histopathology. While these studies are not essential for this paper from my perspective, it is essential to get a good description of any overt toxic phenotypes (or lack thereof) and body weight changes.

Response to Reviewers' comments:

Reviewer #1: In this manuscript, utilizing a number of experiments including *in vitro* metabolomics and *in vivo* MRI, the authors demonstrate that 7ACC2 is a potent MPC inhibitor that appears to not target MCT1. The authors also find that 7ACC2 reduces tumor spheroid growth and radiosensitizes tumor cells *in vivo*. The discovery of a new MPC inhibitor is already an interesting development and they do this very well. However, they do not do an adequate job of connecting this drug mechanism data to the phenotypic effects. All drugs have off-target effects and one needs to be very careful when attempting to conclude mechanistic information from phenotypic data. It is possible that 7ACC2 is an MPC inhibitor, which is quite clearly the case, and that its effects on tumor growth are due to another off-target mechanism.

The comments of the Reviewer are well taken and we had at heart to address each of them in our revised manuscript.

Therefore, the following experiments should be conducted:

- 1. Treat spheroids with UK5099, to see if this pharmacological MPC inhibitor also reduces tumor spheroid growth and induces glucose uptake.*
- 2. Genetically knockout MPC in the tumor spheroids, to determine if MPC depletion also reduces its growth and induces glucose uptake.*

In suppl. Fig. 5E-F, we present new data documenting the stimulatory effects of the *bona fide* MPC inhibitor UK5099 on glucose uptake. The growth inhibitory effects of UK5099 on tumor spheroids is also documented in Suppl. Figure 5G. Both sets of data are in agreement with the effects of 7ACC2 reported in Fig. 5.

As suggested by the Reviewer, in a new set of experiments, we have silenced MPC1 in tumor cells used to generate spheroids (Fig. 5J) and confirmed a significant increase in glucose uptake as probed with 2DG-IR (Figure 5K-L). In this model of MPC1 KO, however, no effect on tumor growth could be detected (see Suppl. Fig. 5G) contrary to the acute treatment with either 7ACC2 (Fig. 5A-B) or UK5099 (Suppl. Fig. 5G), suggesting a compensatory mechanism when MPC is completely silenced before the initiation of spheroids; this observation is briefly discussed.

- 3. Treat xenografts in nude mice with UK5099, to test whether MPC inhibition by UK5099 also radiosensitizes tumor cells and causes tumor reoxygenation.*
- 4. Introduce MPC KO xenografts in nude mice, to see if KO MPC radiosensitizes tumor cells and causes tumor reoxygenation as well.*

To abide by the Reviewer's comments, we have now performed new series of *in vivo* data using either MPC KO tumor cells or the UK5099 compound. With two distinct clones of MPC1-shRNA-expressing cancer cells, we confirmed a net radiosensitizing effect resulting from MPC1 gene silencing (Suppl. Fig. 6A). Tumor growth inhibitory effects resulting from irradiation were also

significantly increased in mice treated with UK5099 (Suppl. Fig. 6B); a significant decrease in pimonidazole staining reflecting tumor reoxygenation was also measured (Suppl. Fig. 6C) thereby confirming the reduction in tumor hypoxia observed upon 7ACC2 exposure (Fig. 6A-B).

We thank the Reviewer for suggesting these experiments which, we agree, altogether reinforce our conclusions by further proving the relationship between inhibition of mitochondrial pyruvate transport and tumor growth inhibitory effects including via radiosensitization.

Reviewer #2: *The present manuscript provides extensive indirect evidence that 7ACC2 inhibits mitochondrial pyruvate transport. It further provides evidence that 7ACC2 can inhibit xenograft growth and sensitize such tumors to radiotherapy. Overall, I found the quality of the evidence good, and the metabolic reasoning straightforward, transparent, and accurate. Particularly pleasing is the use of multiple measurement modalities, at the 2D culture, 3D culture, and xenograft level, to reach generally consistent conclusions.*

We thank the Reviewer for the nice comments. Our manuscript has now been revised to include additional informations as suggested by the Reviewer.

Suggested improvements before publication:

1. *The authors should carefully think through the difference between "mitochondrial pyruvate transport" inhibition (in title, and well shown) vs. "MCP inhibition" which implies a specific molecular target where direct evidence is limited. Wording should be chosen to accurately reflect the nature of the data and to include caveats where data is limited.*

The text was modified to reflect the inhibition of mitochondrial pyruvate transport or the inhibition of MPC activity (instead of direct MPC inhibition) since –we agree– we do not have molecular evidence of the direct interaction between MPC and 7ACC2. Of note, we have now also performed experiments using (i) MPC1-silenced cancer cells and (ii) the *bona fide* MPC inhibitor UK5099 and confirmed the increased glucose uptake and radiosensitizing effects observed with 7ACC2 (new Figs. 5J-L, Suppl. Figs. 5E-G and Suppl. Figs 6A-C).

2. *The first part of the title seems unnecessary. Lactate-based metabolic symbiosis may or may not exist. The contribution of this paper is to show that 7ACC2 blocks mitochondrial pyruvate transport and thereby inhibits tumor growth and sensitizes to radiotherapy.*

We understand the comment of the Reviewer and we have edited the title to remove the reference to lactate-based symbiosis. We have kept the reference to the uptake of lactate since the originality of our findings also arises from the demonstration that *extracellular* lactate uptake may be inhibited by blocking *mitochondrial* pyruvate uptake. During the revision of our manuscript, two papers published in *Nature* and *Cell* (Faubert et al. 2017; Hui et al., 2017) have emphasized the role of

lactate as a prominent nutrient to support oxidative metabolism in cancer cells. These data further support the need for anticancer drugs able to block lactate uptake (and lactate-based symbiosis). Although MCT inhibitors could appear as obvious drug candidates for this purpose, our manuscript provides several sets of data that support the use of inhibitors of mitochondrial pyruvate transport (over MCT inhibitors).

3. The authors should briefly discuss alternative MCP inhibitors, their state of development, pros and cons, and potency and in vivo utility.

We now discuss recently identified compounds able to block mitochondrial pyruvate transport (page 13). Initially developed for other indications than cancer, these drugs are associated with well-known adverse effects.

4. The authors should provide the careful documentation of the safety/toxicology of the employed doses of 7ACC2 in vivo. One would anticipate that blocking mitochondrial pyruvate uptake could result in substantial toxicity in almost any organ, perhaps especially the brain or in prolonged exercise. Evidence of blood brain barrier penetration (or not) would be useful, as would exercise studies and tissue histopathology. While these studies are not essential for this paper from my perspective, it is essential to get a good description of any overt toxic phenotypes (or lack thereof) and body weight changes.

The comment of the Reviewer is well taken. We now provide additional information on the 7ACC2 safety and pharmacokinetics issues in the Discussion (page 13) as well as Supplementary information. Briefly, in our attempts to evaluate different routes of administration, we found that besides intraperitoneal injection (with DMSO as solvent, our current data), i.v. (NMP/HPβcyclodextrin) and oral (Tween 80 + Carboxymethylcellulose) administrations were achievable with similar plasma half-life (4.5, 2.8 and 3.9 hours, respectively); 7ACC2 was also shown to be blood brain barrier-permeant (plasma half-life = 5.0 hours) with a brain/plasma distribution ratio = 1.24. Importantly, regardless of the route of administration, at the dose employed in our *in vivo* experiments (3 mg/kg), we did not observe neither significant weight loss nor abnormal lesions on gross necropsy.

We have now also performed experiments where mice were exercised to exhaustion (as judged by refusal to remain on the treadmill belt). As anticipated, while 7ACC2-treated mice at rest did not show signs of disturbance, these mice showed a reduced run capacity when compared to sham-treated mice (Suppl. Fig. 6D). This side effect could certainly represent a discomfort for cancer patients exposed to MPC inhibitors but should not prevent the use of such therapeutic modality as long the patient is kept at rest.

REVIEWERS' COMMENTS:

Reviewer #1 (Remarks to the Author):

The authors have addressed our concerns and the manuscript is now appropriate for publication in Nature Communications.

Reviewer #2 (Remarks to the Author):

This is a strong revision of an already strong paper. I have a few small suggestions prior to publication:

1. The media used in some figures, including Fig 3A and Fig 5 is not clear. Please specify in the text or figure legend.

2. While it is fortunately easy to find 7ACC2 structure by Google, it would still be good to put structure in paper to avoid any confusion in the future and probably to add one sentence describing where 7ACC2 came from. Similarly, a few words of introduction to 7ACC2 would be useful in the abstract, e.g. "we identified 7ACC2, a small molecule previously described as an MCT inhibitor"

3. (optional) While I understand that the authors believe "Lactate exchange between glycolytic and oxidative cancer cells optimizes tumor growth," I don't personally consider this established fact. I would be more comfortable if the paper started with "It has been proposed that lactate exchange". The authors may want to consider related changes to the introduction that reflect the possibility that lactate exchange between glycolytic and oxidative cancer cells is less important than lactate exchange between the tumor and the circulation.

Response to Reviewers' comments:

Reviewer #1 (Remarks to the Author):

The authors have addressed our concerns and the manuscript is now appropriate for publication in Nature Communications.

Reviewer #2 (Remarks to the Author):

This is a strong revision of an already strong paper. I have a few small suggestions prior to publication:

1. The media used in some figures, including Fig 3A and Fig 5 is not clear. Please specify in the text or figure legend.

Information about media is now more clearly provided in the Method section (in particular, under the subheadings "Gas Chromatography-Mass Spectrometry (GC-MS) analysis of metabolites" for Figure 3A and "3D cultures" for Figure 5).

2. While it is fortunately easy to find 7ACC2 structure by Google, it would still be good to put structure in paper to avoid any confusion in the future and probably to add one sentence describing where 7ACC2 came from. Similarly, a few words of introduction to 7ACC2 would be useful in the abstract, e.g. "we identified 7ACC2, a small molecule previously described as an MCT inhibitor"

The structure of 7ACC2 has been added as Supplementary information. About the second part of the Reviewer's comment, it is true that some companies interpreted our original finding of 7ACC2 being able to block lactate uptake as evidence that 7ACC2 was a *bona fide* MCT inhibitor. However, we had originally claimed that the failure to block lactate efflux with 7ACC2 was making the mechanism of action of this compound uncertain. We strongly believe that now writing that this compound had previously been described as a MCT inhibitor would further maintain the confusion, wrongly advertising this compound as a dual blocker of MCT and MPC. The *Introduction* was thus edited to read that "7ACC2, an anticancer compound originally reported to block lactate influx but not efflux" with references to our two original papers (refs 21 and 22); a similar description of the status of 7ACC2 at the beginning of this study is made in the *Results* (page 5) and *Discussion* (end of page 11) sections.

3. (optional) While I understand that the authors believe "Lactate exchange between glycolytic and oxidative cancer cells optimizes tumor growth," I don't personally consider this established fact. I would be more comfortable if the paper started with "It has been proposed that lactate exchange". The authors may want to consider related changes to the introduction that reflect the possibility that lactate exchange between glycolytic and oxidative cancer cells is less important than lactate exchange between the tumor and the circulation.

The *Abstract* was modified to read: "Lactate exchange between glycolytic and oxidative cancer cells *is proposed to* optimize tumor growth."

The *Introduction* was modified to read: "these studies also validate that **blood-borne (and not only tumor-derived) lactate** may fuel oxidative cancer cells."